# WarmServe: Enabling One-for-Many GPU Prewarming for Multi-LLM Serving

Chiheng Lou [1]   Sheng Qi [1]   Rui Kang [2]   Yong Zhang [2]   Chen Sun [2]   Pengcheng Wang [2]   Xuanzhe Liu [1]   Xin Jin [1]

## Abstract

Deploying multiple models within shared GPU clusters is a key strategy to improve resource efficiency in large language model (LLM) serving. Existing multi-LLM serving systems improve GPU utilization at the cost of degraded inference performance, particularly time-to-first-token (TTFT). We attribute this degradation to the lack of awareness regarding future workload characteristics. In contrast, recent analyses have shown the strong periodicity and long-term predictability of real-world LLM serving workloads. In this paper, we propose *one-for-many GPU prewarming*, which proactively loads parameters from multiple models onto GPUs based on workload forecasts. These prewarmed weights enable the system to promptly instantiate serving instances upon encountering request bursts. We design and implement WarmServe, a multi-LLM serving system incorporating three key techniques: (1) a model placement algorithm that optimizes prewarming decisions to minimize cross-model prewarming interference, (2) a KV cache reservation strategy that repurposes idle KV cache space on running GPUs for prewarming new models, and (3) an efficient GPU memory switching mechanism for tensor management. Evaluation on real-world datasets shows that WarmServe reduces tail TTFT by up to $50.8\times$ compared to the state-of-the-art autoscaling-based system, while supporting up to $2.5\times$ higher request throughput than the GPU-sharing system.

## 1. Introduction

The rapid evolution of large language models (LLMs) has led to a diverse ecosystem of specialized models tailored for tasks such as chat (OpenAI, 2023a; 2025a; DeepSeek-AI, 2025b; Grattafiori et al., 2024), coding (Anthropic, 2025b; Team, 2025), and reasoning (DeepSeek-AI, 2025a; OpenAI, 2023b; Google, 2025). This proliferation introduces significant challenges for model serving, as platforms must now concurrently host a multitude of model variants to meet varied user demands. For example, OpenAI offers more than 20 model variants through its API (OpenAI, 2025b).

Efficiently serving multiple LLMs is challenging due to highly dynamic workloads. Since request volumes fluctuate frequently (Zhang et al., 2025; Duan et al., 2024; Yu et al., 2025c), a dedicated allocation strategy (i.e., a set of GPUs is dedicated to serving a particular model) leads to severe GPU under-utilization during idle periods. To solve this problem, existing solutions can be divided into two categories: (1) autoscaling approaches that scale the number of instances for each model according to current loads (Fu et al., 2024; Zhang et al., 2025; Yu et al., 2025b; Lou et al., 2026; Yu et al., 2025a; Hu et al., 2025; Zhu et al., 2025; Liu et al., 2025; Mei et al., 2026), and (2) GPU sharing approaches that colocate multiple models on the same GPU through spatial and temporal sharing (Li et al., 2023; Duan et al., 2024; Yu et al., 2025c; Patke et al., 2024; Gao et al., 2025; Xiang et al., 2025).

Unfortunately, although these approaches can improve cluster-wide GPU utilization, they impose a non-negligible impact on inference performance. Autoscaling suffers from significant cold-start latency, as initializing new instances on-demand during request bursts is time-consuming. Conversely, GPU sharing avoids initialization delays but severely constricts the KV cache capacity available to each model—a critical resource for maintaining high throughput and handling long sequences in LLM serving.

We attribute the limitations of existing approaches to the lack of awareness of future workload characteristics. Due to this lack of foresight, autoscaling is only triggered after bursty requests arrive, and the model colocation strategy must be stable over time. However, recent production traces reveal a key insight: while short-term request arrivals are inherently stochastic, the long-term statistical trends of LLM workloads exhibit strong periodicity (Wang et al., 2025; Stojkovic et al., 2025; Xiang et al., 2026). Our analysis (Section 2.2) confirms this, demonstrating that peak

[1]School of Computer Science, Peking University [2]Huawei Technologies Co., Ltd. Correspondence to: Xin Jin <xinjinpku@pku.edu.cn>, Chen Sun <sunchen48@huawei.com>.

*Proceedings of the $43^{rd}$ International Conference on Machine Learning*, Seoul, South Korea. PMLR 306, 2026. Copyright 2026 by the author(s).

LLM request volumes within 5-minute windows can be predicted with an average relative error of 7.3%.

Leveraging this high predictability, we propose a proactive autoscaling approach centered on *LLM prewarming* to bridge the gap between resource efficiency and inference performance. Instead of reacting to request arrivals, the system provisions model replicas in anticipation of forecasted load spikes. When the predicted future demand of a specific model exceeds its current capacity, the system proactively launches backup replicas on idle GPUs, enabling the cluster to directly serve potential bursts without delay.

Nevertheless, maintaining exclusive backup replicas for every model is resource-intensive and restricts prewarming scalability. To address this, we introduce *one-for-many GPU prewarming*, which loads parameters from multiple models into the memory of a single GPU. Once a specific model encounters a request burst, it can immediately instantiate an active serving instance using its pre-loaded weights. The system then evicts non-target parameters to guarantee exclusive GPU access for the active instance. Since evicting irrelevant weights is substantially faster than loading weights on demand, one-for-many GPU prewarming improves inference performance while reducing the GPU resources required for prewarming.

While prewarming has been extensively adopted to reduce cold start latency for serverless functions (Sahraei et al., 2023; Yu et al., 2024; Du et al., 2020; Wei et al., 2023), adapting this technique to LLMs introduces two unique challenges:

1. **Cross-model prewarming interference**. LLMs are often distributed across multiple GPUs (e.g., via tensor parallelism). A prewarmed model is only functional if **all** its constituent GPUs are available. Consequently, allocating a single GPU to one model evicts and invalidates multiple other prewarmed models that reside on this GPU, leading to cross-model prewarming interference that complicates placement decisions.
2. **Transient prewarming windows**. LLM workloads are extremely volatile. For example, LLM requests can surge 5× within two seconds in production (Zhang et al., 2025). This leaves a narrow window to prepare new models. Given the massive size of LLM checkpoints, most prewarming attempts fail to complete before the next allocation cycle, resulting in wasted effort and failed prewarming.

To address these challenges, we present WarmServe, a multi-LLM serving system designed to unleash the potential of one-for-many GPU prewarming. To mitigate cross-model interference, WarmServe formulates the model placement problem and employs a prewarming algorithm that maxi-

mizes the prewarming efficacy under potential evictions by strategically isolating high-priority models.

To guarantee that prewarming succeeds within transient windows, WarmServe proactively initiates prewarming before a GPU is officially released. We observe that GPUs approaching release typically have idle memory as their active load falls below peak capacity. Leveraging this slack, WarmServe loads new parameters into the unused KV cache space of these still-active GPUs. Once released, these GPUs can quickly transition into a serving instance of the pre-loaded models, regardless of when reallocation occurs. We develop a strategy to compute the required KV cache for ongoing requests on these GPUs, and introduce a GPU memory switching mechanism to efficiently manage diverse tensors.

Experiments on real-world datasets show that WarmServe reduces the tail TTFT by up to 50.8× compared to the state-of-the-art autoscaling-based system, and is capable of handling up to 2.5× more requests compared to the GPU-sharing system.

In summary, we make the following contributions.

- We identify the potential of model prewarming in multi-LLM serving and introduce one-for-many GPU prewarming to efficiently prepare backup model replicas based on future workload forecasts.
- We formulate the joint problem of model placement and proactive prewarming, developing an optimized placement algorithm alongside a dynamic KV cache reservation strategy. We also propose a GPU memory switching mechanism for efficient tensor management.
- We conduct a comprehensive evaluation of WarmServe, demonstrating its superior performance and efficiency compared to state-of-the-art solutions.

## 2. Background and Motivation

In this section, we introduce LLM serving and analyze the long-term predictability of real-world LLM serving traces.

### 2.1. LLM Serving

LLM serving is the end-to-end process wherein a client sends a request, known as a *prompt*, to a serving engine, which in turn performs inference and streams the response back to the client. User experience is primarily measured by two key metrics: time-to-first-token (TTFT) and time-per-output-token (TPOT). TTFT is the latency to generate the first token, while TPOT represents the average time interval for generating each subsequent token.

**LLM Inference**. LLM inference consists of two stages. The *prefill* stage generates the first token from the initial prompt. During this stage, the key and value vectors for each token in

the prompt are computed and stored in GPUs, known as the *KV cache*. The *decoding* stage then generates the remaining outputs autoregressively. In each decoding step, the model computes and stores the key and value vectors of the last token in the sequence, and generates a new token. As LLM checkpoints are large, they are typically deployed across multiple GPUs using model parallelism, which partitions the weights and synchronizes results at specific stages during inference. To improve GPU utilization, modern serving systems process multiple requests in a batch. A maximum batch size is typically configured to prevent performance degradation when too many requests are processed.

**Multi-LLM Serving**. Large-scale LLM service providers, such as OpenAI (OpenAI, 2025b), Google (Google, 2025), and Anthropic (Anthropic, 2025a), offer a catalog of multiple models tailored to different use cases. Consequently, the serving cluster must be capable of concurrently hosting these models and efficiently managing dynamic workloads for each. In this case, a static GPU allocation strategy leads to significant under-utilization of GPUs since LLMs have fluctuating real-time loads. To address this inefficiency, a variety of multi-LLM serving systems have emerged, which fall into two categories:

• *Autoscaling solutions* that dynamically control the number of instances for each model based on current loads. Upon a load spike, the system creates new instances to serve arriving requests. As instance creation is on the critical path of request handling, these systems reduce creation latency by caching models locally (Fu et al., 2024; Yu et al., 2025a; Hu et al., 2025; Zhu et al., 2025; Liu et al., 2025), fetching models from peers (Zhang et al., 2025; Yu et al., 2025b), and distributing models across servers (Lou et al., 2026). Despite these efforts, bursty requests still suffer from long waiting latency in these systems because instances are created on demand.

• *GPU sharing solutions* that share GPUs across models (Li et al., 2023; Duan et al., 2024; Yu et al., 2025c; Patke et al., 2024; Gao et al., 2025; Xiang et al., 2025). Models are initially deployed and colocated on the same GPUs, and are allocated specific ratios of GPU computational power based on the traffic. In these systems, GPU sharing limits the KV cache space of each model. Additionally, these systems usually increase the degree of parallelism for LLMs to colocate more models, which degrades inference performance.

WarmServe adopts the autoscaling approach, i.e., it assigns dedicated GPUs to instances and scales the instances for each model. However, unlike existing autoscaling systems that passively create instances after bursty requests arrive, WarmServe *prewarms* models in advance by loading their parameters into shared idle GPUs. This synthesis of on-demand autoscaling and shared GPU preparation allows WarmServe to harness the key benefits of each design.

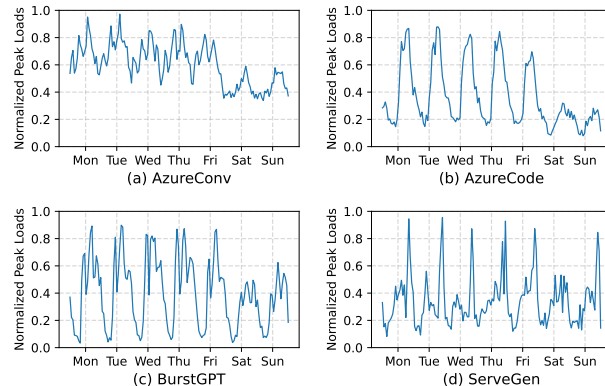

*Figure 1.* Normalized peak loads in 5-minute windows for different traces. Data smoothed using cubic spline interpolation.

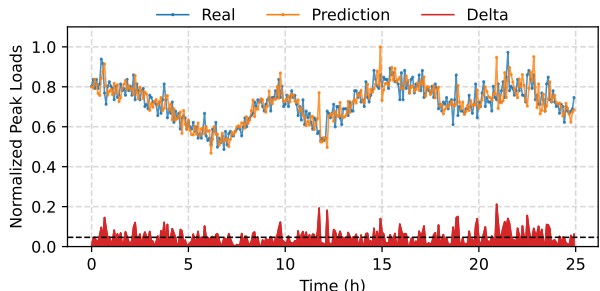

*Figure 2.* Normalized real and predicted peak loads in 5-minute windows for the AzureConv (Stojkovic et al., 2025) trace. The delta represents the absolute error between predicted and ground-truth loads. The black dotted line indicates the average value of the delta.

### 2.2. Workload Predictability

While the short-term burstiness of LLM requests is often regarded as unpredictable (Li et al., 2023; Weng et al., 2022), recent analyses of real-world LLM services have revealed that the long-term statistical characteristics of requests are relatively periodic and predictable (Wang et al., 2025; Stojkovic et al., 2025; Xiang et al., 2026). In this paper, we focus on the characteristics of the average and peak loads, which refer to the average and maximum number of concurrent requests within a specific time window, respectively.

We validate workload predictability by taking peak load as a representative case. Our analysis leverages production traces including AzureConv (Stojkovic et al., 2025), AzureCode (Stojkovic et al., 2025), BurstGPT (Wang et al., 2025), and ServeGen[1] (Xiang et al., 2026). Figure 1 shows the values of load peaks in 5-minute windows for different traces. The results demonstrate that the load peaks of a single LLM follow a periodic pattern in all traces, indicat-

---

[1]We choose the **m-small** workload as it is the most popular model.

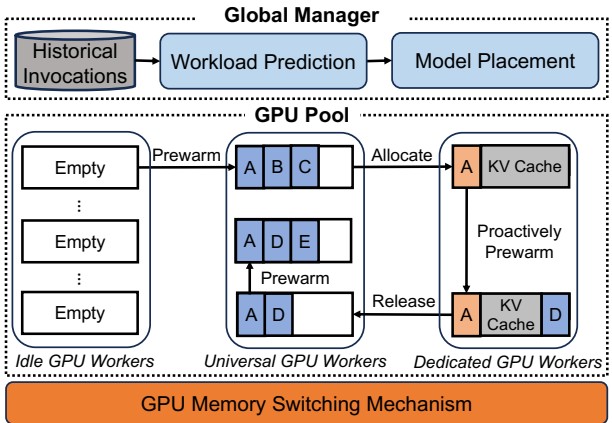

*Figure 3.* WarmServe system architecture.

ing that peak values within a given time window can be effectively predicted using historical data.

We further develop a simple corrective seasonal predictor to predict the peak load for each time window (Section 3.1). Figure 2 shows the prediction performance of our predictor on AzureConv. By using the recorded peak loads of both previous days and recent windows, the predictor achieves an average accuracy of 92.7%, which is sufficient for model prewarming. Notably, predicting the average load is even easier, with an average accuracy of 94.7%. The detailed results for all traces are shown in Section 4.6.

This paper leverages the per-model workload predictability to prewarm appropriate instances for different LLMs. By ensuring that sufficient instances are prewarmed for each LLM, most incoming requests can be served without delay. Additionally, we propose one-for-many GPU prewarming to prepare multiple LLMs simultaneously, significantly reducing the GPU resources consumed by prewarming.

## 3. Method

We design and build WarmServe, a multi-LLM serving system that performs one-for-many GPU prewarming based on workload predictions. Figure 3 shows the architecture of WarmServe. WarmServe classifies GPU workers into three types: idle, universal, and dedicated. For idle GPU workers, WarmServe prewarms several LLMs on them by loading model weights onto GPU memory, converting these workers into universal GPU workers. Prewarming a model often requires multiple GPUs, each holding part of the weights. When one of the prewarmed models encounters request bursts, the loaded model weights enable the GPUs to quickly create engines and start serving, becoming dedicated GPU workers. Prewarmed weights of other models on these GPUs are evicted. WarmServe also allows prewarming to happen

on dedicated GPU workers by storing weights in their unused KV cache space.

At runtime, WarmServe predicts the future workload for each LLM (Section 3.1), and leverages a model placement algorithm (Section 3.2) to generate prewarming actions. For dedicated GPUs that will be released soon due to decreased loads, WarmServe proactively prewarms new models onto their unused KV cache so that the GPUs can directly become universal GPU workers after being released (Section 3.3). Finally, WarmServe adopts a GPU memory switching mechanism to manage GPU memory (Section 3.4), ensuring the flexible coexistence of serving model parameters, prewarmed model weights, and KV cache.

WarmServe avoids degrading steady-state inference performance by ensuring exclusive GPU access for serving instances. Meanwhile, it adopts the idea from GPU sharing systems (Li et al., 2023; Duan et al., 2024; Yu et al., 2025c; Patke et al., 2024; Gao et al., 2025; Xiang et al., 2025) to colocate prewarmed models on universal GPU workers. This approach maximizes prewarming efficiency and significantly reduces the TTFT for cold-start models since the model preparation process is launched in advance.

### 3.1. Workload Prediction

**Problem Formulation**. We partition time into windows of length $t$ minutes. The workload predictor aims to forecast, for each model, the average and peak load in the upcoming time window given historical invocations.

**Input**. Taking peak load as an example, the historical invocations are denoted as $L_{s,j}$, which represents the maximum concurrent requests of the LLM in the $j$-th time window on day $s$.

**Output**. The predicted peak load for the $i$-th window on day $k$, denoted as $\hat{L}_{k,i}$.

**Prediction Algorithm**. WarmServe predicts the workload for the next time window using a simple corrective seasonal predictor (CSP) (Holt, 2004; Winters, 1960; Ghosh et al., 2009). The prediction is based on two components: a seasonal pattern and a corrective pattern. For the seasonal pattern, we use the average peak load of the $i$-th window in the past several days, calculated as follows.

$$P_{k,i} = \frac{1}{D} \sum_{d=1}^{D} L_{k-d,i}, \qquad (1)$$

where $P_{k,i}$ is the seasonal term for day $k$, window $i$. $D$ is the number of historical days to sample, typically set to 7.

For the corrective pattern, we utilize recently observed peak loads to quantify the deviation between the current trend

and the historical seasonal pattern, calculated as follows.

$$\Delta_{k,i} = \frac{\sum_{w=1}^{\min(i,N)}(L_{k,i-w} - P_{k,i-w}) \cdot 2^{w-1}}{2^{\min(i,N)} - 1}, \quad (2)$$

where $\Delta_{k,i}$ is the corrective term for day $k$, window $i$. $N$ is the size of the lookback window, typically set to 10. This weighting scheme gives more importance to recent errors.

The final prediction, $\hat{L}_{k,i}$, is the sum of the seasonal component and this corrective term.

$$\hat{L}_{k,i} = P_{k,i} + \Delta_{k,i}. \quad (3)$$

The prediction of average load follows the same procedure. Due to the periodic nature of LLM workloads, our predictor demonstrates high performance on real-world datasets. With a 5-minute window size, it achieves an average relative error of 5.3% for average loads and 7.3% for peak loads on the AzureConv (Stojkovic et al., 2025) trace. While more sophisticated prediction algorithms, such as ARIMA (Box et al., 2015) and deep learning models (Ni et al., 2026; Liu et al., 2017), could potentially yield higher accuracy, we found that CSP is sufficiently effective for guiding model prewarming while imposing negligible overhead.

### 3.2. Model Placement

**Problem Formulation**. Based on prediction results, WarmServe prewarms models on GPUs that have available memory. During this process, we need to determine which model to prewarm and where to store its weights.

**Input**. Consider the cluster has $U$ GPUs, where the $j$-th GPU has $M_j$ free memory. Assume we have $N$ models in total, where the $i$-th model already has $K_i$ active serving instances. For each model, the predictor will give the predicted average load $L_{A_i}$ and peak load $L_{P_i}$ in the next time window. Each model also has characteristics including model size $S_i$, required number of GPUs $D_i$, and maximum batch size $B_i$.

**Output**. A list of models $\{m_1, \cdots, m_k\}$ to prewarm and the set of GPUs $\{P_1, \cdots, P_k\}$ to place each model in. Note that the model list can be duplicated since a model can have multiple prewarmed replicas to prepare for burst requests.

**Placement Algorithm**. For the $i$-th model, we want to prewarm at most $\lceil L_{P_i}/B_i \rceil - K_i$ replicas to handle future requests. WarmServe first computes a *prewarming score* for each replica that represents its necessity. The score is determined by (1) the gap between the number of active instances and the predicted future loads and (2) the time cost to launch an instance from scratch.

Based on prewarming scores, WarmServe sorts replicas by their scores in descending order, then iterates through replicas and attempts to place each one. For each replica, we first

---

**Algorithm 1** Prewarming Model Placement Algorithm

**Input:** #models $N$, #GPUs $U$; model size $S_i$, parallelism degree $D_i$, batch size $B_i$, current number of instances $K_i$, predicted average and peak load in the next time window $L_{A_i}, L_{P_i}$, cold start latency $T_{c_i}$ for each model $i$; available memory $M_j$ for each GPU $j$.
**Output:** to-prewarm models $\{m_1, \cdots, m_k\}$ and placement $\{P_1, \cdots, P_k\}$.

  $Replicas, PrewarmList \leftarrow \emptyset$
  **for** $i \in \{1, 2, \cdots, N\}$ **do**
    **for** $r \in \{1, 2, \cdots, \lceil L_{P_i}/B_i \rceil - K_i\}$ **do**
      $score_{i,r} \leftarrow$ Prewarming score for the $r$-th replica of model $i$
      $Replicas \leftarrow Replicas \cup (i, score_{i,r})$
    **end for**
  **end for**
  $Replicas' \leftarrow$ **sort**($Replicas$, key = score, order = descending)
  **for** $(i, score) \in Replicas'$ **do**
    $G \leftarrow$ GPU groups consisting of $D_i$ GPUs, each with at least $\lceil S_i/D_i \rceil$ free memory
    **if** $G$ is $\emptyset$ **then**
      **continue**
    **else**
      $H_g, R_g \leftarrow$ Highest and sum of scores of replicas that share GPUs with each GPU group $g \in G$
      **if** $\min\{H_g\} < score$ **then**
        $g' \leftarrow g \in G$ that has least $R_g$ while $H_g < score$
      **else**
        $g' \leftarrow g \in G$ that has least $R_g$
      **end if**
    **end if**
    $PrewarmList \leftarrow PrewarmList \cup (i, g')$
    $M_j \leftarrow M_j - \lceil S_i/D_i \rceil, \ j \in g'$
  **end for**
  **return** $PrewarmList$

---

derive several GPU groups that can serve as candidate placement positions, and then greedily select the optimal one. The selection process tries to isolate high-score replicas to prevent mutual interference and place lower-score replicas in a way that minimizes their impact on existing replicas. Specifically, we prioritize groups where the new replica's score is higher than that of any other existing replica nested within the group. If multiple such groups are available, the one with the minimum sum of scores from its nested replicas is chosen. Otherwise, the algorithm defaults to selecting the group with the minimum sum of scores. Algorithm 1 outlines the model placement algorithm. For a more detailed explanation, please refer to Appendix B.

## 3.3. Proactive Prewarming

WarmServe leverages proactive prewarming to quickly prewarm models under dynamic LLM workloads. Typically, an LLM serving cluster adopts an autoscaler to scale the number of serving instances for each model. If the GPU utilization of a model falls below a predefined threshold, the autoscaler attempts to shut down some of its instances. At this time, WarmServe proactively loads weights from different models into the GPUs of these to-be-terminated instances. While they continue processing existing requests, we utilize their unused KV cache space to store new parameters, as shown in Figure 3. These instances have plenty of unused KV cache space since they are usually underutilized and no longer receive additional requests.

**Problem Formulation**. Dedicated GPUs that will be released in the near future should inform the global manager how much KV cache space they can provide to store new models. As these GPUs are still processing ongoing requests, they must determine how much KV cache space to reserve for existing requests.

**Input**. Denote $Q$ as the number of existing requests and $B$ as the maximum batch size of the instance. Assume the total KV cache capacity is $C$ and $T$ is the size of currently consumed space.

**Output**. The amount of KV cache space to be reserved for existing requests, denoted as $R$.

**Reservation Strategy**. We consider the reservation problem with two targets: (1) the expected KV cache usage based on the number of ongoing requests and (2) the current KV cache usage plus a reserved buffer for potential KV space expansion. The reserved KV cache capacity is computed as the maximum of two targets:

$$R = \max\left( C \cdot \frac{Q}{B},\ T + \frac{C}{B} \right). \tag{4}$$

Since generation lengths are unpredictable, we reserve extra KV cache for ongoing requests using the estimated average per-request usage $C/B$. When the reserved KV cache space becomes insufficient to serve existing requests, we evict prewarmed weights to free up memory.

## 3.4. GPU Memory Management

WarmServe uses a memory switching mechanism to efficiently manage GPU memory among different prewarmed LLMs. The mechanism leverages the CUDA Virtual Memory Management (VMM) API (NVIDIA, 2025a) to manipulate the GPU page table, similar to previous works (Yu et al., 2025c; Cheng et al., 2026; Prabhu et al., 2025). This API allows us to pre-allocate all physical pages in the GPU and map virtual memory addresses to them on demand.

*Table 1.* Specification of models in our experiments.

| Model | Size (GB) | # Required GPUs | Trace Sampling Day |
|-------|-----------|-----------------|--------------------|
| Llama2-7B-0 | 12.55 | 1 | Thursday |
| Llama2-7B-1 | 12.55 | 1 | Friday |
| Llama2-13B | 24.24 | 2 | Saturday |
| Llama2-70B | 128.49 | 4 | Sunday |

*Table 2.* TTFT of models in different scenarios.

| Model | TTFT (s) | | |
|-------|----------|--------------|------|
| | No Prewarm. | With Prewarm. | Warm |
| Llama2-7B | 25.30 | 0.32 (79.06× ↓) | 0.05 (506.0× ↓) |
| Llama2-13B | 28.04 | 0.32 (87.63× ↓) | 0.05 (560.8× ↓) |
| Llama2-70B | 40.29 | 0.67 (60.13× ↓) | 0.05 (805.8× ↓) |

Our method begins by pre-allocating multiple virtual memory regions on each GPU, where the size of each region equals the total available GPU memory. Each of these regions serves as a *prewarm slot* that is used to store an individual prewarmed model. During the prewarming phase, we assign models different slots and map physical pages to them. Models can load their weights into corresponding slots. When a prewarmed model needs to occupy the GPU and become a serving instance, we map physical KV cache space to its prewarm slot and invalidate all other prewarm slots. The serving engine will use this prewarm slot to perform all computations and only access the serving model. In this way, a universal GPU can switch between different prewarmed models upon load spikes.

All page table modifications are overlapped with other operations so that the overhead of our GPU memory switching mechanism is negligible. More details are in Appendix C.

**Implementation**. WarmServe is implemented based on vLLM (Kwon et al., 2023). We use Ray Actor (Moritz et al., 2018) to instantiate GPU workers and perform an end-to-end prewarming during serving engine creation. The code of WarmServe is open-source and is publicly available at https://github.com/LLMServe/WarmServe. For more implementation details, please refer to Appendix D.

## 4. Evaluation

In this section, we present experimental results to validate the efficiency and effectiveness of WarmServe. We also perform ablation studies to verify the effectiveness of individual components. For additional experimental results, please refer to Appendix A.

### 4.1. Experimental Setup

**Testbed.** We evaluate WarmServe on two GPU servers, each with eight GPUs. A single GPU has around 1K TFLOPS of

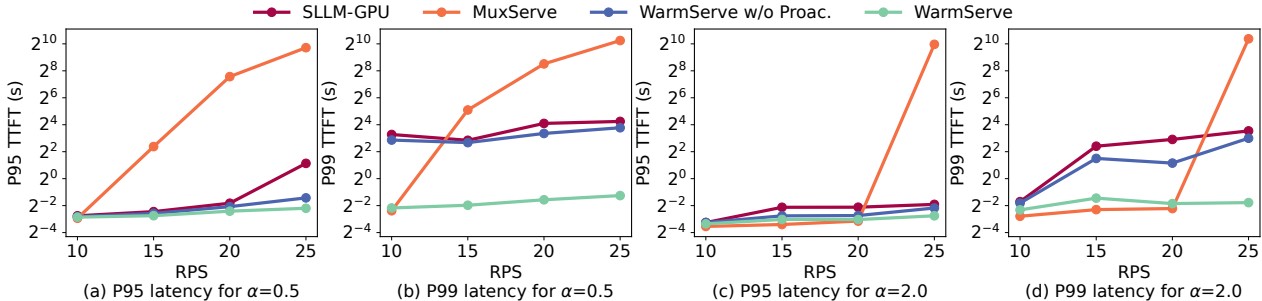

*Figure 4.* TTFT of systems in different settings. A logarithmic scale is used for the y-axis.

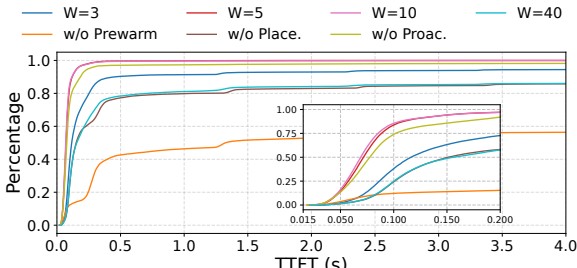

*Figure 5.* TTFT CDF comparing WarmServe variants. The plot shows the impact of (1) varying prediction window sizes and (2) removing one of the techniques: model prewarming, the model placement algorithm, and the proactive prewarming strategy. When disabling our model placement algorithm, the fallback strategy is round robin.

computational power for dense FP16 computation and intra-server GPUs are connected via 400 GB/s NVLink. Each server is equipped with eight 200 Gbps RDMA NICs for inter-server connectivity. Model weights are stored in host memory and are loaded into GPUs on demand via PCIe 5.0 x16 channels.

**Metrics.** We primarily focus on the time-to-first-token (TTFT) in the experiments, defined as the end-to-end duration from request submission to receipt of the first token. We also provide experimental results regarding the time-per-output-token (TPOT) in the appendix.

**Workloads.** Following a similar approach to previous works (Fu et al., 2024; Duan et al., 2024; Lou et al., 2026; Xiang et al., 2025), we use the Llama2 model series (Meta, 2023) and duplicate Llama2-7B to increase the scaling frequency. The details of models are shown in Table 1. With respect to memory capacity, a single GPU can store the weights of six whole Llama2-7B models.

Following prior work (Zhang et al., 2025), we generate workloads from the AzureConv trace (Stojkovic et al., 2025), sampling model requests from different days in the trace. Per-model request rates follow a power-law distribution with

exponent $\alpha$, as in (Duan et al., 2024). We vary the global requests-per-second (RPS) to control the total load.

**Baselines.** We compare WarmServe with the following baselines.

• **ServerlessLLM-GPU (SLLM-GPU)** (Fu et al., 2024): Since model weights already exist in host memory, we extend the caching mechanism of ServerlessLLM to GPUs. When an instance stops, its model parameters remain in GPU memory for future invocation.

• **MuxServe** (Duan et al., 2024): It colocates models on GPUs and uses CUDA MPS (NVIDIA, 2025b) to isolate inference tasks. LLM serving instances are created in advance. As the original MuxServe implementation is based on an old vLLM version, we reimplement it using the same vLLM version as WarmServe for fairness.

All systems use vLLM (Kwon et al., 2023) as the inference backend. SLLM-GPU and WarmServe adopt a batch size of 32, while MuxServe uses the configuration generated by itself in each setting. We do not compare with BlitzScale (Zhang et al., 2025) since its documentation is incomplete and we are unable to install it in our environment.

### 4.2. Prewarming Effectiveness

We first evaluate the effectiveness of prewarming by measuring the TTFT of models under different scenarios. Table 2 shows the TTFT of models under (1) *No Prewarming*, (2) *With Prewarming*, and (3) *Warm*. In the first two cases, the serving engine is created upon request arrival, and model weights are previously stored in host memory and GPU memory, respectively. In the *Warm* case, the engine is already running and can process requests immediately.

The results demonstrate that successful prewarming can significantly reduce TTFT. With prewarming, WarmServe can achieve a 60.13×–87.63× TTFT reduction. In this case, the first token can be generated in ∼670ms for a Llama2-70B model when the serving engine is created on demand.

This is sufficient to satisfy most service level objectives for LLM serving in production.

### 4.3. End-to-End Experiments

We further evaluate the effectiveness of WarmServe through end-to-end experiments. Figure 4 presents the tail TTFT of systems under various scenarios. To validate the effectiveness of proactive prewarming, we also conduct experiments on WarmServe with proactive prewarming disabled.

The results show that WarmServe consistently delivers low TTFT across all settings. Compared to SLLM-GPU, WarmServe achieves a $1.07\times$–$10.06\times$ reduction in P95 TTFT and a $1.53\times$–$50.79\times$ reduction in P99 TTFT by rapidly launching new instances from prewarmed models. Moreover, due to the dynamic nature of workloads, proactive prewarming drastically improves prewarming efficiency, reducing tail TTFT by $1.03\times$–$32.87\times$. The tail TTFT in autoscaling-based systems remains relatively stable, as higher traffic only increases the frequency of scaling rather than the percentage of requests that experience delays.

The GPU-sharing system, MuxServe, exhibits performance comparable to WarmServe under light loads due to its pre-created model instances. However, its static model placement strategy limits serving capacity and degrades performance for colocated models, leading to severe queuing under heavy loads. For example, at an RPS of 15 and $\alpha$=0.5, MuxServe's P95 and P99 TTFT are $34.5\times$ and $134.0\times$ higher than those of WarmServe, respectively. Overall, WarmServe can handle up to $2.5\times$ more requests than MuxServe while maintaining low TTFT.

### 4.4. Ablation Study

We conduct an ablation study to analyze the impact of the prediction window size and validate the effectiveness of our key components: model prewarming, the model placement algorithm, and the proactive prewarming strategy. We set RPS to 25 and use a 5-minute window size by default.

Figure 5 shows that disabling model prewarming, the model placement algorithm, or the proactive prewarming strategy reduces the percentage of requests meeting the 100ms TTFT threshold to $0.15\times$, $0.29\times$, and $0.88\times$ of the baseline, respectively. These results highlight that while the prewarming mechanism itself is fundamental to performance, our model placement algorithm and proactive prewarming strategy provide a critical additional boost.

The prediction window size involves a critical trade-off: small windows (e.g., 3 min) suffer from prediction instability, while overly large windows (e.g., 40 min) fail to capture transient load dynamics. These settings serve only $0.46\times$ and $0.30\times$ as many requests within the 100ms threshold compared to the 5-minute baseline. Notably, even with sub-

*Table 3.* P99 TTFT (s) of systems under 512-GPU simulation.

| RPS | WarmServe | SLLM-GPU | MuxServe | Pred. Autoscal. |
|-----|-----------|----------|----------|-----------------|
| 320 | 0.18 | 5.1 ($28.3\times \uparrow$) | 0.27 ($1.5\times \uparrow$) | 0.18 ($1.0\times \uparrow$) |
| 480 | 0.25 | 5.1 ($20.4\times \uparrow$) | 35 ($140\times \uparrow$) | 0.25 ($1.0\times \uparrow$) |
| 640 | 0.40 | 12 ($30.0\times \uparrow$) | 400 ($1000\times \uparrow$) | 1.84 ($4.6\times \uparrow$) |
| 800 | 0.38 | 8.9 ($23.4\times \uparrow$) | 1300 ($3400\times \uparrow$) | 4.42 ($11.6\times \uparrow$) |

optimal windows, WarmServe consistently outperforms the no-prewarming baseline, demonstrating the robustness of our GPU prewarming mechanism.

### 4.5. Large-Scale Simulation

We conduct large-scale simulations to further evaluate the scalability of WarmServe. The simulation is based on the same workload generation method as in the end-to-end experiments ($\alpha = 0.5$), but with a larger cluster of 512 GPUs. To show the necessity of prewarming, we also include a baseline that uses the same workload predictor as WarmServe to perform predictive autoscaling without prewarming.

The simulation results are shown in Table 3. WarmServe consistently achieves low tail TTFT across all load levels, demonstrating its high scalability. Conversely, the performance of SLLM-GPU and MuxServe degrades significantly as the load increases, with TTFT increasing by up to $30\times$ and $3400\times$, respectively. Predictive autoscaling can achieve the same TTFT as WarmServe under light loads through timely autoscaling. However, as load increases, the frequency of scaling events rises, leading to more requests experiencing scaling delays. Owing to the one-for-many GPU prewarming mechanism, WarmServe can leverage limited spare resources to prepare for future demands and maintain low TTFT even under heavy loads.

### 4.6. Workload Prediction

We evaluate our workload predictor, CSP, by predicting workload characteristics for a single model using production traces including AzureConv, AzureCode, BurstGPT, and ServeGen. We utilize the first week of data from each trace, partitioned into 5-minute windows. For every window between Tuesday and Sunday, CSP predicts the average and peak loads based on the historical data of all preceding windows. Figure 6 shows the hourly prediction performance across these traces. Throughout the evaluation period, the predicted loads align closely with the ground truth, demonstrating the strong long-term predictability inherent in production LLM traces.

## 5. Related Work

**Multi-LLM Serving.** Existing systems generally follow two paradigms: *autoscaling*, which allocates exclusive

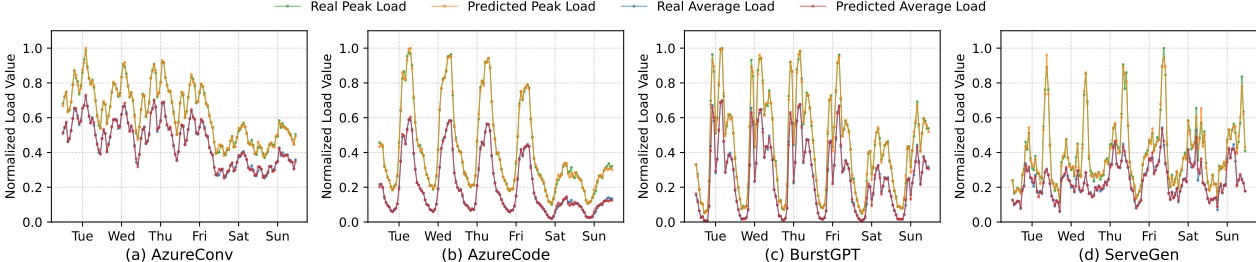

*Figure 6.* Hourly averages of 5-minute-window real and predicted loads.

GPUs to instances (Fu et al., 2024; Yu et al., 2025a; Hu et al., 2025; Zhang et al., 2025; Yu et al., 2025b; Zhu et al., 2025; Liu et al., 2025; Lou et al., 2026; Mei et al., 2026), and *GPU sharing*, which colocates multiple models to increase utilization (Li et al., 2023; Duan et al., 2024; Yu et al., 2025c; Patke et al., 2024; Gao et al., 2025; Xiang et al., 2025). P-LoRA (Ni et al., 2026) prefetches LoRA adapters. However, its scope is limited to lightweight LoRA modules rather than full-parameter models. Autoscaling approaches often optimize model loading via caching (Fu et al., 2024; Yu et al., 2025a; Hu et al., 2025; Zhu et al., 2025; Liu et al., 2025), high-bandwidth transfers (Zhang et al., 2025; Yu et al., 2025b), or distributed fetching (Lou et al., 2026). In contrast, WarmServe adopts a proactive autoscaling approach and prewarms models using predictions.

**Serverless Prewarming.** Prewarming has been widely used to reduce cold start latency in serverless computing (Bhasi et al., 2021; Gunasekaran et al., 2020; Stojkovic et al., 2023; Cai et al., 2024; Shahrad et al., 2020; Sahraei et al., 2023; Yu et al., 2024; Du et al., 2020). However, LLMs present unique challenges: they span multiple GPUs and require full weight loading. WarmServe tailors prewarming placement and memory management for LLM serving.

**KV Cache Management.** Efficient KV cache management is critical for LLM performance. While prior works optimize KV cache management via paged memory (Kwon et al., 2023), long-sequence handling (Lin et al., 2024; Wu et al., 2024; 2025), or offloading (Qin et al., 2025; Sheng et al., 2023), WarmServe repurposes unused KV cache memory to store prewarmed model weights, enabling a GPU worker to transition roles near-instantaneously.

## 6. Conclusion

This paper presents WarmServe, a multi-LLM serving system that leverages the long-term predictability of LLM workloads to enable one-for-many GPU prewarming. WarmServe employs a specialized model placement strategy to mitigate cross-model prewarming interference, and proactively loads models onto active GPUs to speed up prewarming. Eval-

uation results show that WarmServe drastically improves inference performance compared to existing systems.

## Acknowledgments

We sincerely thank the anonymous reviewers for their valuable feedback on this paper. This work was supported in part by the National Key Research and Development Program of China under Grant 2022YFB4500700, the Scientific Research Innovation Capability Support Project for Young Faculty under Grant ZYGXQNJSKYCXNLZCXM-I1, and the National Natural Science Foundation of China under Grant 62172008 and 62325201. Xin Jin and Chen Sun are the corresponding authors. Chiheng Lou, Sheng Qi, Xuanzhe Liu, and Xin Jin are also with the Key Laboratory of High Confidence Software Technologies (Peking University), Ministry of Education.

## Impact Statement

This paper presents work whose goal is to advance the field of Large Language Model Serving Systems. There are many potential societal consequences of our work, none which we feel must be specifically highlighted here.

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

# A. Additional Experimental Results

## A.1. TTFT for Different Models

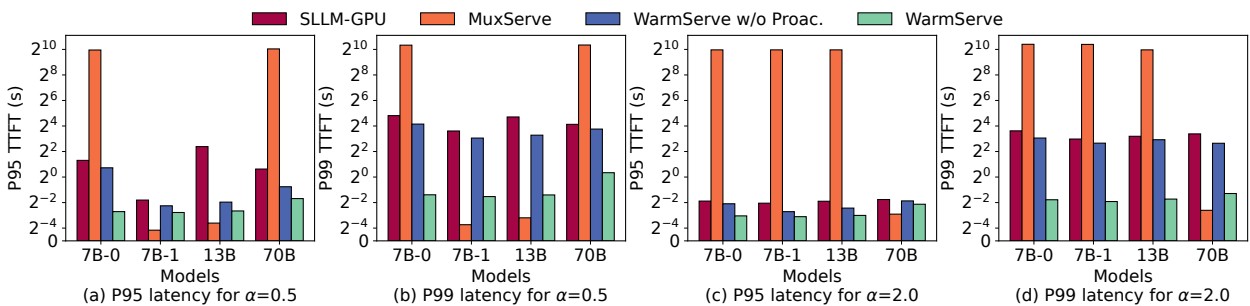

*Figure 7.* TTFT for models under RPS=25. A logarithmic scale is used for the y-axis.

Figure 7 details the TTFT for different models under RPS=25 in our experiments. For $\alpha$=0.5, the MuxServe-chosen placement colocates the 7B-0 and 70B models on 8 GPUs, whereas for $\alpha = 2.0$, the 7B-0, 7B-1, and 13B models all share the same 8 GPUs. Although MuxServe can achieve low TTFT for models that do not share GPUs, colocated models are constrained by limited GPU resources, leading to severe request queuing. GPU sharing also introduces performance interference. For example, under the $\alpha$=2.0 setting, the 13B model receives only 8.2% of total requests but still experiences significant queuing. This is due to resource contention from the high-demand 7B-0 model, which processes 70.2% of the requests.

In contrast, WarmServe maintains stable TTFT performance across all models and settings. It achieves a $1.29\times$–$73.60\times$ reduction in tail TTFT compared to SLLM-GPU, and a $1.20\times$–$46.52\times$ reduction compared to itself with proactive prewarming disabled.

## A.2. TPOT for Different Systems

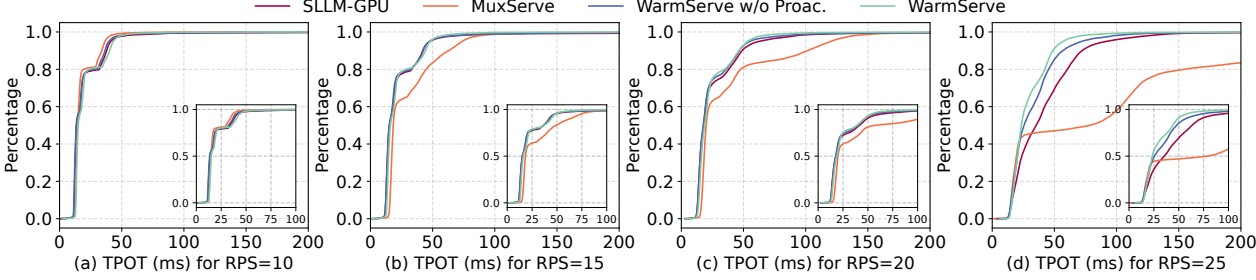

*Figure 8.* TPOT CDF of systems under $\alpha$=0.5.

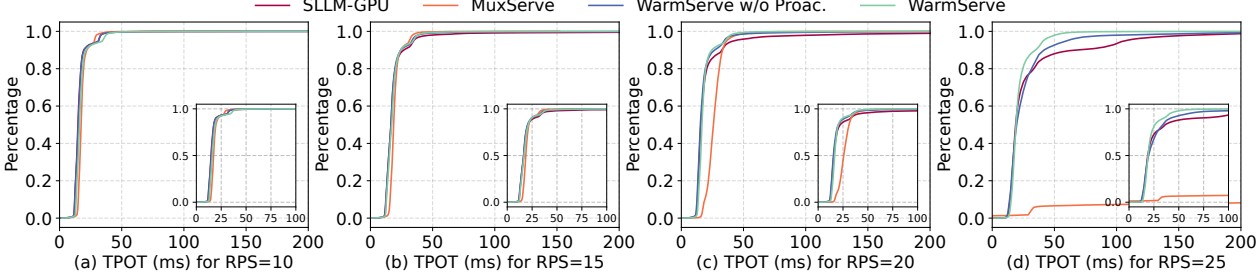

*Figure 9.* TPOT CDF of systems under $\alpha$=2.0.

We provide the TPOT CDF of systems in Figure 8 and Figure 9. MuxServe's strategy of increasing model parallelism and colocating multiple models on the same GPUs leads to severe inference performance degradation. For example, under heavy load (RPS=25) in Figure 8, while over 60% of requests in autoscaling-based systems achieve a TPOT under 50ms, the

overhead from GPU sharing causes over 40% of MuxServe's requests to exceed a TPOT of 100ms. This analysis highlights the fundamental advantage of WarmServe: its prewarming techniques drastically reduce TTFT, and its exclusive allocation of GPU workers preserves inference performance.

### A.3. Evaluation on AzureCode Dataset

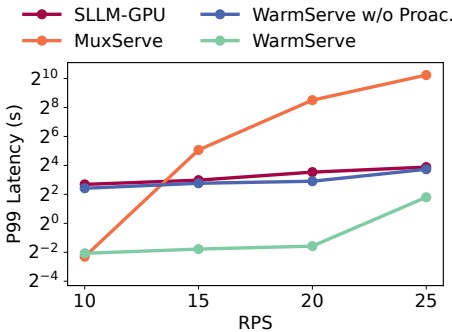
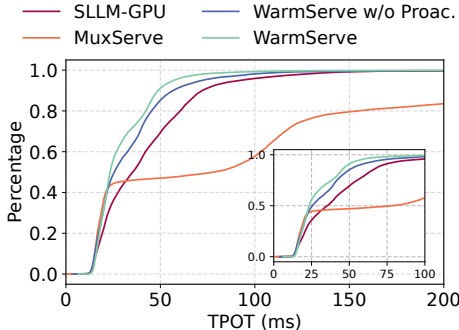

*Figure 10.* Tail TTFT latency on AzureCode for $\alpha$=0.5.  *Figure 11.* TPOT CDF on AzureCode for $\alpha$=0.5 and RPS=25.

Figure 10 and Figure 11 depict the TTFT and TPOT of systems on the AzureCode trace, with $\alpha$ fixed at 0.5. This trace contains fewer requests than AzureConv, making it less predictable. Consequently, WarmServe exhibits a slight performance degradation on this trace.

Figure 10 shows that, on the AzureCode trace, WarmServe achieves a 4.23×–34.52× reduction in P99 TTFT over SLLM-GPU, and a 3.81×–23.34× reduction compared to itself with proactive prewarming disabled. Under a light load (RPS=10), MuxServe's P99 TTFT is slightly lower (0.85× that of WarmServe). However, this marginal benefit disappears under heavier loads, where MuxServe's latency becomes significantly higher. Furthermore, MuxServe fails to provide consistent performance guarantees, incurring an average TPOT that is 3.26× higher than that of WarmServe, as shown in Figure 11.

### A.4. Prewarming Hit Ratio

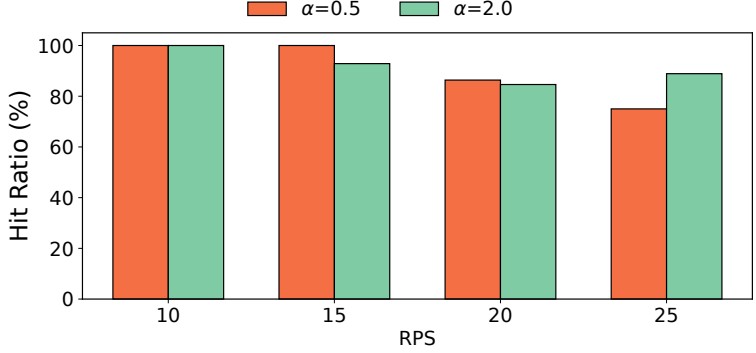

*Figure 12.* Prewarming hit ratios in end-to-end experiments.

Figure 12 shows the effectiveness of one-for-many prewarming by presenting the prewarming hit ratios of WarmServe. Under light loads, the cluster has sufficient idle GPUs, allowing WarmServe to prewarm all required model replicas and ensure a 100% prewarming hit ratio. As the load increases, the resources available for prewarming are reduced. Despite this, WarmServe successfully prewarms the majority of required instances, achieving an average hit ratio of 82% under RPS=25. This high hit ratio confirms the viability of one-for-many prewarming under pressure.

# B. Model Placement Algorithm

## B.1. Prewarming Score Computation

To compute the score of each prewarming replica, we classify replicas for each model into two categories.

• *Basic replicas.* These replicas are prewarmed to ensure sufficient model instances under average loads. For models whose active instances cannot meet the predicted average load, we create basic replicas to fill these gaps.

• *Burst replicas.* These replicas are used to tackle load spikes. After creating sufficient basic replicas, we continue to prewarm additional replicas until the total number of instances (both active and prewarmed) can serve the predicted peak load. These additional replicas are designated as burst replicas, ensuring enough serving capacity under peak loads.

Formally, consider a model with $K$ active instances and a batch size of $B$, and with predicted average and peak loads $L_A$ and $L_P$, respectively. The number of basic ($N_{basic}$) and burst ($N_{burst}$) replicas to be prewarmed is calculated as follows.

$$N_{basic} = \max\left(\lceil L_A/B \rceil - K,\ 0\right); \tag{5}$$
$$N_{burst} = \max\left(\lceil L_P/B \rceil - N_{basic} - K,\ 0\right). \tag{6}$$

Within each category, replica priority is further refined using a prewarming score, $S$. The score is calculated differently for each replica type.

$$S_{basic} = \exp\left(-\frac{i}{N_{basic} + N_{burst}}\right) \cdot T_c; \tag{7}$$
$$S_{burst} = \exp\left(-\frac{N_{basic} + i}{N_{basic} + N_{burst}}\right) \cdot T_c \cdot \frac{L_P - L_A}{L_A}, \tag{8}$$

where $i$ is the zero-indexed rank of the replica within its category, and $T_c$ is the latency of loading the whole model weights into GPUs, which is obtained through offline profiling.

The score for basic replicas (Eq. 7) is a product of two factors. The first, an exponential decay term, models the diminishing returns of prewarming replicas. As more replicas are prewarmed (increasing $i$), the incremental utility of the next one decreases. The second factor, $T_c$, prioritizes models with longer loading times, as they incur a higher penalty if a prewarmed instance is unavailable.

For burst replicas, the score (Eq. 8) contains an additional burstiness factor, $(L_P - L_A)/L_A$. This term represents the burstiness of the load in the upcoming time window, ensuring that models expecting a larger spike in traffic are prioritized accordingly. Note that we make sure basic replicas always have higher priorities than burst replicas, regardless of their scores.

## B.2. Placement Algorithm

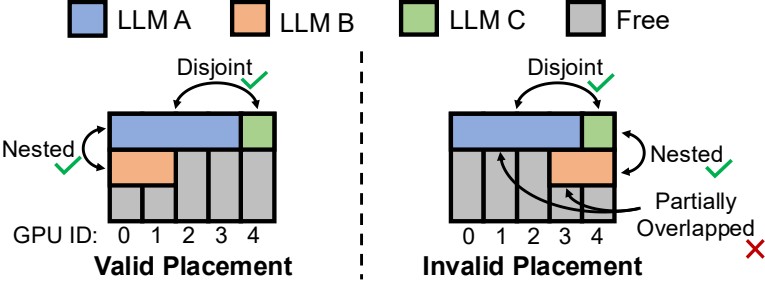

*Figure 13.* Placement guideline of WarmServe.

The placement algorithm of WarmServe is governed by two primary guidelines. The first guideline strictly prohibits partial GPU sharing. Specifically, for any two models, the set of GPUs allocated to them must either be entirely disjoint or one set must be a complete subset of the other. This constraint ensures that prewarming contention for GPU resources is limited to models in a nested arrangement, thereby reducing the interference patterns that arise from more complex overlaps.

Figure 13 provides a visual representation of this placement restriction. In a valid placement, models are either disjoint (using separate GPUs) or nested (where one model's GPUs are a subset of another's). Placements with partial overlap, where models share a subset of GPUs while also holding exclusive ones, are explicitly forbidden. The rationale for this prohibition is that partial overlaps can escalate resource contention. For example, as shown in the figure, moving LLM B from GPUs (0,1) to GPUs (3,4) would cause it to partially overlap with LLM A, and this would simultaneously create a new contention point with LLM C.

The second placement guideline focuses on prioritizing models based on their anticipated demand. WarmServe aims to isolate models that have high prewarming hit probabilities by allocating them to disjoint GPU sets. This strategy prevents these critical, high-priority models from interfering with one another's performance. Subsequently, models with a lower prewarming hit probability can be colocated on these same GPUs, leveraging unused resources with minimal performance impact on the primary models.

After calculating prewarming scores, our placement algorithm strategically allocates prewarming replicas to GPU workers. It begins by calculating the prewarming score for each replica and then proceeds to place them according to specific placement guidelines. Replicas are processed in descending order of their prewarming scores, with basic replicas prioritized over burst replicas. For each replica $r$, the following procedure is executed.

• *Candidate worker identification*. First, the algorithm identifies a set of candidate GPU workers. This set includes all idle and universal workers in the cluster with sufficient available memory. To facilitate proactive prewarming, dedicated workers in a grace period are also considered candidates. A replica of a model with size $S$ and parallelism degree $D$ requires $S/D$ memory per GPU worker.

• *Placement group formation*. Next, from the pool of candidate GPU workers, the algorithm attempts to form valid placement groups. A valid group must meet two requirements: (1) all workers in the group must be located on the same server to guarantee inference performance, and (2) the group's workers must not partially overlap with any other existing prewarming replica. If no valid group can be formed, we move on to the next replica.

• *Optimal group selection*. If one or more valid groups exist, the algorithm greedily selects the optimal one. The selection process prioritizes groups where the new replica's score is higher than any other existing replica that is nested within the group. If multiple such groups are available, the one with the minimum sum of scores from its nested replicas is chosen. Otherwise, the algorithm defaults to selecting the group with the minimum sum of scores.

## C. GPU Memory Switching Mechanism

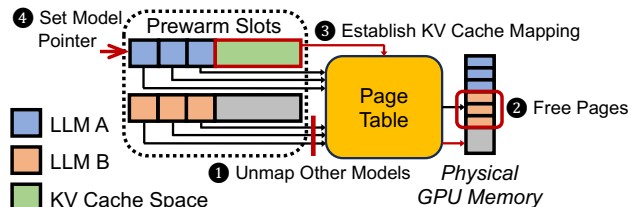

*(a)* Transition a universal GPU worker to a dedicated one.

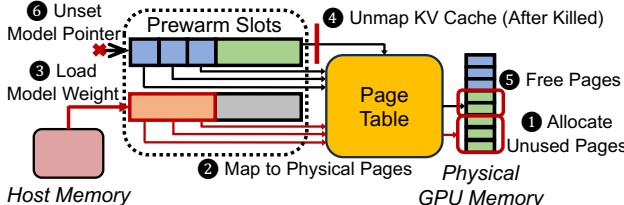

*(b)* Proactively prewarm models on a dedicated GPU worker (1-3) and then seamlessly transition the GPU to a universal GPU worker (4-6).

*Figure 14.* Overview of GPU memory switching mechanism in WarmServe.

Figure 14(a) illustrates the process of converting a universal GPU worker into a dedicated one for a specific model. Upon a successful prewarm hit, we first release the resources of other prewarmed models by unmapping their virtual pages and freeing the underlying physical pages. Next, we create additional mappings for the prewarm slot containing the target model. The unmapped virtual pages within this slot are mapped to all remaining physical pages on the GPU, which will serve as the KV cache. At this point, a complete one-to-one mapping is established between the virtual address space of the prewarm slot and the entire physical GPU memory. Finally, we direct the inference framework to this active slot by configuring the model pointer. This ensures that the framework accesses the model weights and KV cache through a single, contiguous virtual address space, effectively concealing the non-contiguous nature of the underlying physical pages and isolating it from other inactive prewarm slots.

This mechanism is also applicable when launching an instance with a model that has not been prewarmed. In this case, a series of initialization steps is performed. First, all existing prewarmed slots are reclaimed and the loaded model weights are invalidated. Next, an empty slot is allocated to the target model, and all physical pages are mapped to it. Finally, the model weights are loaded into the newly allocated slot, making the GPU ready for inference.

Figure 14(b) further illustrates how WarmServe proactively prewarms models on dedicated GPU workers. During a GPU's grace period, unused KV cache blocks are leveraged to prewarm new models. We compute the corresponding physical pages of these blocks, and map them to the prewarm slot of the new model. The model's weights are then loaded into these pages by accessing the slot's virtual address.

When the original instance terminates, its associated KV cache is reclaimed by unmapping the virtual addresses and freeing the physical pages. The model pointer is also cleared, signaling the absence of an active model. The GPU then transitions back to a universal state, now holding the prewarm slots of both the newly prewarmed models and the previously active one, ensuring readiness for immediate deployment.

**Achieving Near-Zero Switching Overhead**. The primary performance bottleneck in our GPU management lifecycle is the latency of modifying page tables through the CUDA VMM API. For instance, mapping a 10GB virtual address space can take up to 0.2 seconds (Prabhu et al., 2025). To eliminate this overhead, WarmServe decouples page table manipulations from the critical execution path by overlapping them with other operations. These modifications are categorized into two types: mappings for model loading and mappings for the KV cache.

For model loading, WarmServe pipelines the mapping and data transfer operations. As soon as a virtual page is mapped, a data copy for the corresponding weights is immediately triggered. Since the time to map a single page is significantly shorter than the data transfer time, this fine-grained pipelining strategy effectively hides the mapping latency.

For the KV cache, mappings are performed in the background. Since the inference framework consumes cache space at a slower rate than the mapping process produces it, the mapping overhead is fully overlapped. Unmapping operations are also executed asynchronously, as they do not block any subsequent actions.

Consequently, by strategically overlapping all page table manipulations with data transfers and other non-blocking operations, WarmServe ensures the memory switching process incurs negligible overhead.

# D. Implementation

WarmServe is implemented based on vLLM (Kwon et al., 2023), extended with approximately 1.8K lines of C++ and Python code to support universal GPU workers. The global manager is implemented in about 4K lines of Python code.

**GPU Worker Management**. Each GPU worker in the cluster is managed by a Ray Actor (Moritz et al., 2018) that performs model inference. To transition a universal GPU worker into a dedicated one, we instantiate a vLLM serving engine and connect it to actors of its allocated workers. We intercept the engine's remote function calls to these actors to leverage prewarmed model parameters and enable proactive prewarming.

**Prewarming Other Instance Startup Stages**. Creating an instance involves multiple stages apart from loading model parameters (Lou et al., 2026). WarmServe prewarms time-consuming stages to achieve sub-second instance startup, targeting the loading of libraries and establishment of communication groups.

• *Pre-loading library*. We maintain a pool of vLLM processes with all necessary libraries loaded. A new serving engine starts by taking over a prepared process and receiving model-specific arguments. The idle processes are blocked and consume no CPU resources.

• *Pre-establishing communication group*. When a model is prewarmed on multiple GPUs, a communication group is pre-established among them. Upon prewarming hit, the GPU workers leverage this existing group to synchronize messages during inference. We utilize PyTorch MultiWorld (Lee et al., 2024) to enable a GPU runtime to simultaneously host multiple communication groups with different peers.

