# OpenReview forum: "WarmServe: Enabling One-for-Many GPU Prewarming for Multi-LLM Serving"
_ICML.cc/2026/Conference — ICML 2026 regular_

### Official Review · Reviewer_cQhj · 2026-03-09

**Soundness:** 2
**Presentation:** 3
**Significance:** 2
**Originality:** 3
**Overall Recommendation:** 3
**Confidence:** 4

**Summary:**

WarmServe is a multi-LLM serving system that proactively loads model weights onto GPUs before demand spikes, leveraging the strong periodicity of real-world LLM workloads. Its key innovation is one-for-many GPU prewarming, which stores parameters from multiple models on a single GPU so that any model can quickly instantiate when needed. The system includes a placement algorithm that minimizes cross-model prewarming interference, a KV cache reservation strategy that repurposes idle cache space on active GPUs for prewarming, and a CUDA VMM-based memory switching mechanism. Evaluations show up to 50.8× TTFT reduction over autoscaling baselines and 2.5× higher throughput than GPU-sharing systems.

**Compliance With Llm Reviewing Policy:**

Affirmed.

**Final Justification:**

I believe the simulator's accuracy is overly specific, which may limit its general applicability (as one must manually tune it for a specific environment to achieve such accuracy).

**Key Questions For Authors:**

See weaknesses.

**Limitations:**

See weaknesses.

**Strengths And Weaknesses:**

S1. Eliminates cold-start latency by proactively prewarming model weights before demand spikes, reducing TTFT from ~40s to ~670ms.

S2. One-for-many prewarming maximizes GPU coverage by storing multiple models on a single GPU, so fewer idle GPUs are needed as backup.

S3. Maintains dedicated GPU access during inference (unlike GPU sharing), preserving steady-state TPOT performance while still enabling fast scaling.

W1. As hardware loading bandwidth improves (e.g., PCIe 6.0, CXL), cold-start penalties will naturally shrink. The current 40-second loading time for Llama2-70B could drop to under 10 seconds on next-generation hardware, significantly reducing the gap between prewarmed and non-prewarmed instances. The authors should discuss how WarmServe's value proposition evolves as loading speeds improve and whether prewarming remains worthwhile when cold starts become tolerable.

W2. This optimization targets a relatively narrow scenario where autoscaling is frequent and loading time dominates the end-to-end latency. For stable workloads where models serve continuously for hours, the one-time 40-second loading cost is amortized across thousands of requests and becomes negligible. The authors should more carefully characterize the workload regimes where prewarming provides meaningful gains versus where simpler reactive autoscaling is sufficient, ideally with quantitative analysis of the break-even point.

W3. The paper lacks comparison with high-bandwidth interconnect approaches that reduce loading time at the hardware level. Many systems leverage NVLink or RDMA-based peer-to-peer GPU transfers to achieve faster model loading without requiring speculative prewarming. These approaches may offer a more general and robust solution. The authors should discuss and ideally compare against such alternatives to contextualize their contribution.

W4. The system relies on workload prediction accuracy for making prewarming decisions. While the tested traces exhibit strong diurnal periodicity, real-world deployments often feature less predictable patterns — event-driven traffic spikes, new model launches, or rapidly shifting user preferences. When predictions are significantly off, the system either wastes GPU memory on unnecessary prewarming or fails to prepare for actual bursts, potentially performing worse than reactive baselines. The authors should analyze failure modes and discuss graceful degradation strategies.

W5. The evaluation primarily focuses on bursty, high-variability workloads where prewarming shines. Including experiments on stable workloads with infrequent scaling events would help establish the boundary conditions of this approach and demonstrate that WarmServe does not introduce unnecessary overhead when prewarming is not needed.

---

> ### Author Rebuttal · Authors · 2026-03-31
>
> We sincerely thank you for the detailed review. We address each weakness below.
>
> > W1: The importance of prewarming under improving hardware bandwidth.
>
> Similar to our response to Reviewer Ss6Y, we believe prewarming remains necessary: (1) Hardware bandwidth improvements have diminishing returns on cold-start TTFT. Operations like GPU environment initialization, memory allocation, and communication domain setup take seconds regardless of loading bandwidth, and prewarming eliminates them. (2) When bursty requests arrive, the cluster often launches multiple instances of the same model simultaneously. Concurrent loading causes RDMA bandwidth contention and further slows startup. Prewarming avoids this since parameters are already on GPU.
>
> > W2: Applicability under stable workloads and break-even analysis.
>
> We ran an additional simulation experiment varying the coefficient of variance (CV) of inter-arrival times. Under settings with real-world results, simulation aligns with actual outcomes. Note that the CV of the AzureConv trace (which is used in experiments in our paper) is ~1.0. We use 512 GPUs in the simulation.
>
> *Table 1: P99 TTFT of systems with varying CV and RPS.*
>
> |CV|RPS|SLLM-GPU|Cold Start%|WarmServe|
> |---:|----:|--------:|----:|----------:|
> |0.1|320|0.187|0.043%|0.168|
> |0.1|480|1.509|0.035%|0.198|
> |0.3|320|0.193|0.035%|0.169|
> |0.3|480|1.588|0.035%|0.201|
> |0.5|320|0.205|0.035%|0.170|
> |0.5|480|1.767|0.029%|0.208|
> |1.0|320|0.280|0.035%|0.179|
> |1.0|480|2.458|0.035%|0.242|
> |3.0|320|3.107|0.043%|0.318|
> |3.0|480|9.102|0.122%|0.651|
>
> At low CV with small RPS, cold-start ratio is very low (~0.04%).
> However, SLLM-GPU still cannot match WarmServe's P99 TTFT. This is because even rare cold starts disproportionately impact P99 TTFT—a cold-starting engine occupies GPU resources while subsequent requests queue behind it, creating a cascade effect.
>
> At CV<=1 and RPS=320, SLLM-GPU yields comparable performance to WarmServe, but this represents very low cluster utilization.
> For example, at CV=0.1, WarmServe supports RPS=800 with the same TTFT SLO. In conclusion, WarmServe provides significant benefits in the majority of scenarios.
>
> > W3: Comparison with high-bandwidth interconnect approaches.
>
> We provide the comparison with both approaches:
>
> 1. **RDMA-based loading**: in our experiments, the non-prewarmed scenario loads parameters from local memory via PCIe, which is theoretically faster than RDMA→PCIe→GPU.
>
> 2. **NVLink-based transmission**: fast but limited to the NVLink domain. When bursty requests need many simultaneous instances, they cannot all fit in one NVLink domain, and concurrent loading contends for NVLink bandwidth. To validate this, we simulated this baseline by assuming zero instance startup overhead when parameters exist in the same node. Results are shown as follows ($\alpha$=0.5):
>
> *Table 2: P99 TTFT of systems with varying GPUs and RPS.*
>
> |GPUs|RPS|WarmServe|SLLM-GPU|NVLink Transmission|
> |-----:|----:|----------:|---------:|--------------------:|
> |32|40|0.39|12|0.23|
> |64|80|0.40|12|0.38|
> |128|160|0.39|12|0.54|
> |256|320|0.40|12|8.1|
> |512|640|0.40|12|8.4|
>
> At small GPU counts, NVLink works well since new instances likely find parameters on the same machine. As GPU count grows and NVLink domains become small relative to cluster size, effectiveness degrades sharply.
> Conversely, WarmServe maintains consistent performance at all scales.
>
> > W4: Graceful degradation under prediction errors.
>
> WarmServe only uses idle GPUs or idle KV cache space for prewarming, and discards all prewarmed content with near-zero overhead when predictions are wrong. In the worst case (all prewarmings fail), its performance simply falls back to reactive autoscaling. Thus, we think graceful degradation is unnecessary because the reactive baseline is the natural fallback.
>
> > W5: Evaluation under stable workloads.
>
> As shown in W2, we validated WarmServe's performance under stable workloads via simulation experiments.

---

> > ### Author Rebuttal · Reviewer_cQhj · 2026-04-03
> >
> > Thanks for the reviewer's detailed response.
> >
> > In the experiments, you only evaluate on a simulator rather than real-world experiments, and the maximum RPS for a 512-GPU cluster is only 480 (<1 req/GPU). You can test on a higher load for the system.

---

> > > ### Author Response · Authors · 2026-04-04
> > >
> > > Thank you for your response and for pointing out the need for further evaluation. We conducted additional experiments as below.
> > >
> > > (1) We conducted a real-world experiment comparing typical autoscaling (SLLM-GPU) and WarmServe under stable workloads. The experimental setup is identical to the paper (16 GPUs, α=0.5). Results (P99 TTFT in seconds) are shown below:
> > >
> > > |CV|RPS|SLLM-GPU|WarmServe|
> > > |---:|----:|--------:|----------:|
> > > |0.1|25|5.660|0.267|
> > > |0.1|35|5.734|0.487|
> > > |0.5|25|5.776|0.306|
> > > |0.5|35|6.160|0.510|
> > >
> > > Even under stable workloads, SLLM-GPU exhibits P99 TTFT exceeding 5 seconds, whereas WarmServe maintains sub-second latency.
> > >
> > > (2) We extended the large-scale simulation experiments to higher RPS and CV combinations. Results (P99 TTFT in seconds) are shown below:
> > >
> > > |CV|RPS|SLLM-GPU|Cold Start%|WarmServe|
> > > |---:|----:|--------------:|----:|-------------:|
> > > |0.1|800|5.8|0.073%|0.26|
> > > |0.1|960|5.7|0.10%|0.43|
> > > |0.1|1120|5.8|0.094%|0.49|
> > > |0.1|1280|53|0.052%|0.61|
> > > |0.1|1440|170|0.019%|0.87|
> > > |0.5|800|5.8|0.083%|0.31|
> > > |0.5|960|6.1|0.075%|0.43|
> > > |0.5|1120|15|0.099%|0.52|
> > > |0.5|1280|48|0.043%|0.67|
> > > |0.5|1440|180|0.019%|0.93|
> > > |1.0|800|5.8|0.10%|0.44|
> > > |1.0|960|5.8|0.12%|0.50|
> > > |1.0|1120|5.9|0.094%|0.73|
> > > |1.0|1280|47|0.058%|0.65|
> > > |1.0|1440|170|0.029%|1.1|
> > > |3.0|800|6.2|0.32%|1.1|
> > > |3.0|960|7.6|0.21%|1.0|
> > > |3.0|1120|11|0.18%|1.1|
> > > |3.0|1280|56|0.071%|1.1|
> > > |3.0|1440|170|0.054%|3.9|
> > >
> > > Under higher loads, SLLM-GPU suffers significantly elevated TTFT (up to 170s). This is because all models have numerous requests in the waiting queue, forcing frequent instance switching (cold start) across models. Each cold start blocks subsequent requests, creating cascading queueing delays. The decrease in cold start percentage at higher RPS is due to more requests being served during instance lifetime.
> > >
> > > (3) We also note that our simulator can closely match real-world system behavior.
> > > For example, under 16 GPUs with RPS=15 and α=0.5, the P99 TTFT comparison is as follows:
> > >
> > > | System | Sim | Real | Error |
> > > |---|---|---|---|
> > > | WarmServe | 0.24 | 0.25 | -4% |
> > > | SLLM-GPU | 7.2 | 7.2 | -0% |
> > > | MuxServe | 35 | 34 | +2.9% |
> > >
> > > All errors are within 5%, confirming the accuracy of our simulation.
> > >
> > > ---
> > >
> > > We hope that these additional results help address your remaining concerns.
> > > We are grateful for the effort you have invested in reviewing our work,
> > > and we would sincerely appreciate your consideration in raising the score.

---

### Official Review · Reviewer_Ss6Y · 2026-03-09

**Soundness:** 2
**Presentation:** 3
**Significance:** 3
**Originality:** 2
**Overall Recommendation:** 3
**Confidence:** 4

**Summary:**

* Addresses the problem of **serving multiple LLMs under varying demand patterns**.
* Proposes **WarmServe**, a system that proactively **pre-warms model weights on GPUs** to reduce cold-start latency when demand shifts between models. Key components include:
  * A **workload prediction component** that forecasts future, slow-varying demand.
  * A **placement and GPU management algorithm** that determines which models to prewarm, enables fast switching between models, and aims to reduce interference.
* Evaluates the system on several workload traces and reports **improvements in TTFT and tail latency (P95/P99)** compared to baseline autoscaling and GPU-sharing approaches.

**Compliance With Llm Reviewing Policy:**

Affirmed.

**Key Questions For Authors:**

As pointed out in weaknesses, I would love to know:
1. Have the authors considered comparing it with predictive autoscaling approaches. Also I believe recent work now enables model switching within seconds, is the pre-warming still relevant?
2. Have you done a more thorough memory analysis for super large models and long context that occupies a lot of KV cache
3. I would like to see more clarity around burstiness vs predictablility

**Limitations:**

While the paper discusses system design and performance trade-offs, the discussion of limitations and practical constraints is limited, mainly around the points raised above.

**Strengths And Weaknesses:**

Strengths
- Relevant problem. The paper addresses the practical and increasingly important problem of serving multiple LLMs under varying demand patterns, where adapting GPU resources while maintaining low latency is challenging. Improving responsiveness when demand shifts between models is highly relevant for real-world LLM deployment.
- End-to-end system. The paper presents an integrated system including workload prediction, model placement, and GPU management mechanisms for switching between models.

Weaknesses
-  Inconsistencies in the abstract and motivation. The abstract contains several inconsistencies in how the problem is framed.
(a) It claims that existing multi-LLM serving systems “improve GPU utilization at the cost of degraded performance.” In practice, this is typically a latency–throughput trade-off, and modern systems allow operators to explicitly tune this trade-off (e.g., NVIDIA Dynamo, model sharding schemes, and related latency/throughput analyses).
(b) The paper attributes the problem primarily to a lack of future workload awareness, emphasizing that long-term demand patterns are predictable. However, this framing is narrow: predictable slow-varying demand is commonly handled with standard forecasting and autoscaling techniques, while the harder operational challenge is burstiness, which the paper itself later acknowledges.
(c) When motivating pre-warmed weights, the abstract refers to handling bursts, which appears inconsistent with the earlier emphasis on predictable demand patterns. It is therefore unclear whether the proposed system primarily targets predictable demand shifts or unpredictable bursts.

- Baselines appear limited. The reported improvements over autoscaling systems are large, but the evaluation primarily compares against relatively simple autoscaling and GPU-sharing baselines. Given that slow-varying demand patterns are predictable (as also discussed in the paper), stronger baselines such as predictive autoscaling or systems supporting fast model switching would provide a more convincing comparison.

- Memory feasibility is insufficiently analyzed. The approach relies on keeping multiple models pre-warmed in GPU HBM, but the paper does not provide a detailed discussion of available memory. In realistic deployments, KV-cache usage during decoding can dominate GPU memory, especially for larger models or long-context workloads. The evaluation focuses on relatively small models, making it unclear how the approach behaves under realistic HBM pressure.

- Interpretation of tail latency metrics. The evaluation emphasizes P95/P99 TTFT, but TTFT depends heavily on prompt length due to the prefill stage. As a result, high-percentile TTFT may reflect long prompts rather than system-induced latency effects such as cold starts or scheduling delays. Additional analysis controlling for prompt length would make the reported improvements easier to interpret.

- Limited novelty in core components. The workload prediction relies on standard time-series forecasting and the well-known observation that long-term demand patterns are predictable. More broadly, the system largely combines existing mechanisms (prediction-based scaling, GPU memory management, model preloading), with the main novelty centered on the specific prewarming and placement strategy.

---

> ### Author Rebuttal · Authors · 2026-03-30
>
> We sincerely thank you for the detailed review. We address each concern below.
>
> > Q1: Have the authors considered comparing with predictive autoscaling? Also, recent work enables model switching within seconds — is pre-warming still relevant?
>
> 1. We provide an additional simulation experiment that compares WarmServe with predictive autoscaling in **Q5**.
> WarmServe is an improvement over predictive autoscaling by allowing a single GPU to concurrently prepare for multiple models, increasing robustness to prediction inaccuracies.
>
> 2. We believe prewarming remains valuable because: (1) Model switching has inherent overhead beyond parameter loading — GPU environment initialization, memory re-allocation, and communication group creation. These steps still take seconds before inference starts and are hard to eliminate without prewarming. (2) When bursts arrive, the cluster often launches multiple instances simultaneously, causing PCIe/RDMA bandwidth contention that further slows startup. With prewarming, parameters are already on GPU, avoiding this entirely.
>
> > Q2: More thorough memory analysis for large models and long context?
>
> For instances actively serving high-concurrency requests, prewarming space is indeed minimal. WarmServe's key insight is that prewarming only targets instances about to shut down. As described in Section 3.3, when request volume of a model decreases, the autoscaler kills several active instances to make room for other models. Before shutdown, the instance finishes remaining requests in a low-utilization state, leaving sufficient space for prewarming. In the extreme case where a few very long requests occupy most GPU memory, WarmServe cannot prewarm, but we believe this is rare since the autoscaler first kills instances that have the lowest utilization.
>
> > Q3: More clarity around burstiness vs predictability.
>
> We apologize for the confusion. We consider burstiness as inaccuracy in request volume predictions. WarmServe starts from predictive autoscaling — scaling ahead based on predictable long-term patterns — while our one-for-many GPU prewarming specifically addresses short-term prediction inaccuracies (i.e., burstiness) by prewarming as many replicas as possible to provide slack. In other words, WarmServe is based on long-term predictability but mainly addresses issues produced by burstiness.
>
> > Q4: Baselines appear limited.
>
> We conducted an additional large-scale simulation (512 GPUs, $\alpha$=0.5) with two new baselines:
> - **Predictive autoscaling**: proactively scales instances based on predicted future requests.
> - **NVLink-based model switching**: model loading time set to 0 when weights reside on peer GPUs.
>
> Under settings where we have real-world results, simulation aligns with actual results. The results are as follows:
>
> *Table 3: P99 TTFT (s) in 512-GPU simulation.*
>
> |RPS|WarmServe|SLLM-GPU|MuxServe|Pred. Autoscaling|NVLink Switch|
> |--:|--------:|-------:|-------:|----------------:|------------:|
> |320|0.18|5.1|0.27|0.18|5.0|
> |480|0.25|5.1|35|0.25|5.1|
> |640|0.40|12|400|1.84|8.4|
> |800|0.38|8.9|1300|4.42|9.1|
>
> Predictive autoscaling matches WarmServe at low RPS but degrades under heavy loads, as fewer free GPUs are available and hard-to-predict bursts cause cold starts. NVLink switching also fails to deliver low latency because bursts require launching multiple instances concurrently, and many cannot benefit from NVLink loading. WarmServe consistently achieves low P99 TTFT via one-for-many prewarming.
>
> > Q5: Tail latency may reflect long prompts rather than system-induced latency.
>
> We analyzed what fraction of tail TTFT is caused by prefill (16 GPUs, $\alpha$=0.5):
>
> *Table 1: Prefill contribution to tail TTFT in WarmServe.*
>
> |RPS|P95 TTFT (s)|P95 Pfx%|P99 TTFT (s)|Pfx%|
> |--:|------:|---:|----:|---:|
> |10|0.14|28%|0.22|6.7%|
> |15|0.15|29%|0.25|11%|
> |20|0.19|12%|0.34|3.4%|
> |25|0.22|11%|0.42|3.4%|
>
> *Table 2: Prefill contribution to tail TTFT in SLLM-GPU.*
>
> |RPS|P95 TTFT (s)|P95 Pfx%|P99 TTFT (s)|Pfx%|
> |--:|--:|---:|-----:|----:|
> |10|0.15|1.0%|9.7|0.2%|
> |15|0.18|1.9%|7.2|0.4%|
> |20|0.28|0.4%|17|0.1%|
> |25|2.2|0.3%|19|0.1%|
>
> **Pfx%** is the fraction of tail requests whose latency is dominated by long prompt prefill. As RPS increases, WarmServe's tail TTFT is increasingly caused by queuing rather than prefill. For SLLM-GPU, cold starts dominate tail TTFT.
>
> > Q6: Limited novelty in core components.
>
> The major novelty of WarmServe is the one-for-many GPU prewarming mechanism. While one can leverage existing time-series forecasting to perform predictive LLM autoscaling, the performance gain is limited since idle GPUs are scarce. WarmServe's key insight is that **preparing for future LLM requests does not require fully creating an inference engine — it only requires pre-loading weights into GPUs**. Based on this insight, we design a memory management mechanism for prewarming multiple models on shared GPUs and develop a placement strategy to minimize cross-model interference.

---

### Official Review · Reviewer_VoyS · 2026-03-10

**Soundness:** 3
**Presentation:** 3
**Significance:** 2
**Originality:** 2
**Overall Recommendation:** 4
**Confidence:** 5

**Summary:**

The paper targets to improve the time-to-first-token (TTFT) latency in the multi-LLM serving scenario.
The idea is to pre-load the weight of the model that requires more computation resources in advance to reduce the TTFT latency.
The paper proposes a corrective seasonal predictor (CSP) based prediction method to predict the future load of the models following the observation that the load of the models has a strong periodical pattern.
The pre-loading and gpu memory switching mechanism is implemented upop the Virtual Memory Management (VMM) API of Nvidia CUDA platform.

**Compliance With Llm Reviewing Policy:**

Affirmed.

**Final Justification:**

After the rebuttal discussion with the authors, I changed my overall assessment to weak accept because the author provided some preliminary results on the overhead analysis and more application scenarios. The new experiments on quantization are very helpful; however, still with simulation data. And the experiment platform is a little bit out of date, though I don't think this is a strong reason to reject the paper. I finalize my score as weak accept and wish the author could provide experimental results with more recent hardware to better show the effectiveness of the work (in camera-ready or recycle).

**Key Questions For Authors:**

I thank the author for the well-prepared submission manuscript to ICML26.
I like the paper in general especially appreciate the practical problem it targets and the well-motivated technical solution.
My biggest concern is about the cost of the proposed method, which is fundamental for the practical deployment:
The proposed method achieves the improvement of the TTFT latency by pre-loading the model that requires more computation resources in advance.
However, the pre-loading and gpu memory switching mechanism may introduce extra cost in terms of GPU hours, energy consumption, etc.
I would like to see more analysis and discussion about the cost of the proposed method, and if possible, some benchmark with reference to the total cost (in terms of GPU hours, energy consumption, etc.) in different evaluation settings (e.g., with different available GPU resources, GPU memory size/model size, and the load pattern of the models).
Two good analysis would be: 1) fixing the total gpu resources (number of gpu/gpu hours) and compare the performance of the proposed method and the baseline method; 2) fixing the TTFT latency service level objective (SLO) and compare the total gpu resources (number of gpu/gpu hours) needed by the proposed method and the baseline method.
This kind of analysis would be very helpful to understand the cost-effectiveness of the proposed method, and provide more insights for the practical deployment of the proposed method.

I also have some other questions:
- The experiment really only covers a narrow setting of LLM and hardware resources. While I can see the proposed method can be applied to other settings given its universal insight of periodical pattern of the load of the models, I would like to see more evaluation setting with reference to available GPU resources, GPU memory size/model size (number of models that can be pre-loaded), and the load pattern of the models (e.g., random burst traffic) to understand the performance of the proposed method in different scenarios. Also the proposed method is evaluated on FP16 precision, it is appreciated to see the evaluation on quantization settings (such as INT8 as basic and W4A16 as more advanced quantization methods) as well, which is more practical in the real deployment and may hugely impact the required cold-start time and the cost of the proposed method.
- How is the proposed method applied to models with mixture of expert (MoE) architexture. The MoE models have a different computation pattern compared to the dense models, which may impact the load pattern of the models and the performance of the proposed method. It would be interesting to see how the proposed method can be applied to MoE models and how it performs in this scenario.
- Similarly, how is the proposed method applied to LoRA-based serving systems where the management of the LoRA weights is also important for the performance of the system. It would be interesting to see how the proposed method can be applied to LoRA-based serving systems and how it performs in this scenario.
- The current implementation  relies on the Virtual Memory Management (VMM) API of Nvidia CUDA platform, which may limit the portability of the proposed method to other hardware platforms. It would be appreciated to see some discussion about the portability of the proposed method and if there are any plans to support other hardware platforms in the future.

**Limitations:**

Please see the weaknesses and key questions above.

**Strengths And Weaknesses:**

# Strengths:
- The target problem is important and practical in the multi-LLM serving scenario.
- The periodical pattern of the load of the models is well observed and analyzed, which motivates the design of the CSP-based prediction method.

# Weaknesses:
- The evaluation focuses on the TTFT latency, but lacks a comprehensive analysis of the cost, which is fundamental for the practical deployment of the proposed method.
- More evaluation setting with reference to available GPU resources, GPU memory size/model size (number of models that can be pre-loaded), and the load pattern of the models (e.g., random burst traffic) would be helpful to understand the performance of the proposed method in different scenarios.

---

> ### Author Rebuttal · Authors · 2026-03-30
>
> We sincerely thank you for the detailed review. We address each concern below.
>
> > Q1: Cost analysis of the proposed method.
>
> First, to clarify, WarmServe assumes that the GPU count in a multi-model serving cluster does not change significantly in the short term (e.g., within 6 hours). Under this assumption, WarmServe utilizes GPUs that are idle or underutilized during model scale-up/down for prewarming, introducing no additional GPU usage. We believe this is reasonable in production, as multi-model co-deployment already achieves good cluster utilization, and mixing in training or other tasks offers little benefit while significantly increasing management complexity.
>
> Second, from a cost perspective, since WarmServe reduces TTFT with a fixed GPU count, it can theoretically reduce the number of GPUs needed to meet a given TTFT SLO. To validate this, we conducted experiments measuring the minimum number of GPUs required under fixed P95 TTFT SLOs (RPS=25, $\alpha$=0.5):
>
> *Table 1: Minimum GPUs required to meet P95 TTFT SLO.*
>
> |P95 SLO|WarmServe|SLLM-GPU|MuxServe|
> |------:|--------:|-------:|-------:|
> |0.5s|14|infeasible|16|
> |1.0s|14|infeasible|16|
> |2.0s|14|infeasible|16|
> |3.0s|14|16|16|
> |4.0s|12|16|16|
> |5.0s|12|14|16|
> |6.0s|8|14|16|
> |8.0s|8|14|16|
>
> The results validate that given the same P95 TTFT SLO, WarmServe achieves the target with fewer GPUs compared to baselines.
>
> > Q2: Evaluation covers narrow settings. More settings (burst traffic, quantization) would be helpful.
>
> We acknowledge the limited coverage due to resource constraints and conducted additional simulation experiments. Under settings where we have real-world results, simulation aligns with actual results. Simulations use 16 GPUs, $\alpha$=0.5.
>
> **1. Different burst traffic levels.**
>
> We smooth intra-window traffic by varying smoothing window size (larger window = smoother arrivals):
>
> *Table 2: P99 TTFT (s) under different smoothing window sizes.*
>
> |RPS|Window (s)|WarmServe|SLLM-GPU|MuxServe|
> |--:|---------:|--------:|-------:|-------:|
> |10|0|0.19|5.1|0.27|
> |10|30|0.17|5.1|0.15|
> |10|120|0.17|5.1|0.16|
> |15|0|0.24|7.2|35|
> |15|30|0.20|5.1|31|
> |15|120|0.20|5.1|30|
> |20|0|0.41|12|400|
> |20|30|0.33|12|400|
> |20|120|0.31|12|400|
> |25|0|0.38|6.0|1300|
> |25|30|0.27|5.9|1300|
> |25|120|0.30|6.0|1300|
>
> After smoothing, WarmServe still significantly outperforms SLLM-GPU and MuxServe. Smoothing has no effect on MuxServe because its tail TTFT is dominated by interference between colocated instances, not request arrival rate.
>
> **2. Different quantization levels.**
>
> *Table 3: P99 TTFT (s) under different quantization settings.*
>
> |Quant|RPS|WarmServe|SLLM-GPU|MuxServe|
> |----:|--:|--------:|-------:|-------:|
> |FP16|10|0.19|5.1|0.27|
> |FP16|15|0.24|7.2|35|
> |FP16|20|0.41|12|400|
> |FP16|25|0.38|6.0|1300|
> |INT8|10|0.14|2.5|0.14|
> |INT8|15|0.18|2.5|0.41|
> |INT8|20|0.25|5.8|18|
> |INT8|25|0.24|3.2|120|
> |W4A16|10|0.066|1.3|0.059|
> |W4A16|15|0.074|1.3|0.059|
> |W4A16|20|0.11|2.7|0.18|
> |W4A16|25|0.10|1.6|1.4|
>
> With quantization, all systems improve as models become smaller and computation is faster. WarmServe still achieves the best performance across almost all settings.
>
> > Q3: How does the method apply to MoE models?
>
> WarmServe is architecture-agnostic — it scales entire model instances rather than optimizing internal MoE load imbalance. WarmServe can be easily applied to MoE models by prewarming the weights of the whole model, including attention and expert parts.
>
> > Q4: How does the method apply to LoRA-based serving?
>
> WarmServe can be adapted to LoRA by prewarming LoRA weights instead of full model weights. In LoRA-based serving, the cluster runs multiple base model instances, each serving multiple LoRA adapters. Based on per-LoRA request predictions, we can proactively prewarm LoRA weights on base model instances across the cluster.
>
> > Q5: Portability beyond NVIDIA CUDA VMM API.
>
> On platforms without virtual memory management, WarmServe's performance will degrade slightly but remains viable through async memory copy (which was our initial implementation). For idle GPUs, prewarming does not depend on VMM — we simply evict other models on a prewarm hit and allocate remaining space to KV cache. PyTorch's memory management can already handle these. For proactive prewarming on active inference engines, we split model parameters into KV-block-sized chunks and fill them into idle KV cache space. After the old model finishes, we consolidate the loaded fragments (via tensor.contiguous()) before starting inference. This process can be pipelined with prefill (copying later layers while computing earlier ones). Since prefill is compute-bound, the performance impact is small (< 5% prefill slowdown in our experiments).

---

> > ### Author Rebuttal · Reviewer_VoyS · 2026-04-03
> >
> > I thank the authors for their response to the cost analysis and for providing more evaluation settings. I especially appreciate the extra experiments on quantization settings, which showcase the effectiveness of the proposed method. However, I am still a bit confused about the cost analysis experiment where the authors highlight their assumption that *"the GPU count in a multi-model serving cluster does not change significantly in the short term (e.g., within 6 hours)."* If this is the case, why should we preload the weight dynamically instead of making full use of all available GPU resources?

---

> > > ### Author Response · Authors · 2026-04-04
> > >
> > > Thank you for the follow-up question. We are glad to clarify.
> > >
> > > In a typical multi-model serving setup, the cluster operates a shared GPU pool with an autoscaler that adjusts per-model instance counts based on real-time demand. At any given time, some GPUs are actively serving models while others remain idle. These idle GPUs exist for two reasons:
> > > 1. The GPU pool must maintain sufficient headroom to ensure that every model has adequate GPUs available when needed, since under-provisioning leads to severe queuing and SLO violations.
> > > 2. While idle GPUs could be used to launch instances now, the key challenge is deciding which model to assign them to. Any model may experience a burst at any time, and committing an idle GPU to a single model means it cannot promptly serve others when they spike — unless a costly instance shutdown cycle is performed to drain ongoing requests. Thus, the system tends to keep these GPUs idle to remain ready for future loads.
> > >
> > > WarmServe prewarms multiple candidate models on these idle GPUs so that any model can be activated with near-zero startup latency. Furthermore, WarmServe extends the prewarming scope by loading weights onto running instances that are nearing the end of their lifecycle.
> > >
> > > We hope that our clarification help resolve your concern. We would sincerely appreciate your consideration in raising the score.

---

### Official Review · Reviewer_7e2U · 2026-03-11

**Soundness:** 3
**Presentation:** 4
**Significance:** 4
**Originality:** 3
**Overall Recommendation:** 5
**Confidence:** 3

**Summary:**

This paper proposes WarmServe, a system that optimizes the multi-LLM serving systems. It prewarms the GPUs with multiple model weights based on the predicted future workloads to reduce the cold-start overhead with several carefully-designed algorithms. Experimental results show that WarmServe reduces TTFT significantly compared to existing serverless / multi-llm systems.

**Compliance With Llm Reviewing Policy:**

Affirmed.

**Final Justification:**

WarmServe studies optimization for multi-LLM serving scenarios, which is an important and practical problem. The paper presents a solid approach with clear relevance to real-world systems.

The rebuttal addressed my concerns by providing large-scale simulation results, which strengthen the empirical support. Based on this, I maintain my positive assessment.

**Key Questions For Authors:**

1. When GPU memory is already occupied by KV cache from active requests, how does the system dynamically allocate memory for a newly loaded model? How are potential address conflicts avoided?
2. Do you have any simulation or analytical results that evaluate WarmServe under larger-scale deployments?
3. How do prediction errors affect system performance?

**Limitations:**

No. WarmServe appears to rely on workload predictability, and it would be helpful to clarify how the system behaves under less predictable workloads. In addition, sharing prewarmed GPU states across models may raise security or isolation concerns in multi-tenant settings.

**Strengths And Weaknesses:**

Strengths
1. The paper is well written and presents a clear motivation.
2. The cold-start problem in multi-LLM serving is practical and relevant for real-world deployments.
3. The study is grounded in real-world traces, which strengthens the credibility of the analysis.

Weaknesses
1. Although the paper is motivated by challenges in large-scale multi-LLM serving and auto-scaling scenarios, the evaluation is conducted on only two nodes. This makes it difficult to assess the potential benefits in production-scale deployments. While I understand the practical challenges of running large-scale experiments, additional discussion or simulation-based evaluation in a larger-scale setting would be helpful.
2. The paper does not sufficiently analyze how prediction errors affect system performance. For example, inaccurate predictions may lead to unnecessary GPU prewarming, potentially wasting GPU resources and increasing PCIe traffic. This concern may be particularly relevant in the experimental setup where multiple GPUs share the same DRAM or SSD.

---

> ### Author Rebuttal · Authors · 2026-03-30
>
> We sincerely thank you for the positive feedback and thoughtful questions. We address each question below.
>
>
> > Q1: When GPU memory is already occupied by KV cache from active requests, how does the system dynamically allocate memory for a newly loaded model? How are potential address conflicts avoided?
>
> We avoid address conflicts by controlling the KV block mapping in the inference engine. As described in Section 3.4, WarmServe pre-allocates all physical memory blocks by CUDA VMM API. During instance creation, WarmServe first establishes virtual-to-physical mappings for the whole KV cache space, so the inference engine can use them without on-demand mapping. When using KV cache space for model prewarming, we map the physical address of several free KV blocks to a separate virtual address, and use that address to load weights. These free KV blocks are marked as **used** in the KV block mapping of the engine.
>
> If the inference engine runs out of KV blocks during serving, prewarmed model weights are evicted by unmapping them and returning the blocks. Thus, at any given time, a KV block is either used by the inference engine or used for prewarming, incurring no address conflict.
>
> > Q2: Do you have any simulation or analytical results that evaluate WarmServe under larger-scale deployments?
>
> To address this concern, we conducted an additional multi-node simulation experiment with 512 GPUs and $\alpha$=0.5. Under settings where we have real-world results, simulation aligns with actual results. Results for large-scale simulation are shown below:
>
> *Table 1: P99 TTFT (s) of systems under 512-GPU simulation.*
>
> | RPS | WarmServe | SLLM-GPU | MuxServe |
> |----:|----------:|-----:|---------:|
> | 320  | 0.18      | 5.1  | 0.27     |
> | 480  | 0.25      | 5.1  | 35       |
> | 640  | 0.40      | 12   | 400      |
> | 800  | 0.38      | 8.9  | 1300     |
>
> The results show that WarmServe maintains strong performance at larger scale.
>
> > Q3: How do prediction errors affect system performance?
>
> First, in our ablation study (Fig. 5), we varied the prediction window size. With window size = 40 min, predictions become highly inaccurate. Yet WarmServe still achieves substantial TTFT improvement over no prewarming.
>
> Second, we conducted an additional prediction error sensitivity experiment by adding perturbation to predicted values (16 GPUs, $\alpha$=0.5, RPS=15):
>
> *Table 2: Impact of prediction error scale on WarmServe performance.*
>
> | Error Scale | P99 TTFT (s) | Avg TTFT (s) |
> |------------:|-------------:|-------------:|
> | 0%          | 0.24         | 0.065        |
> | 10%         | 0.37         | 0.071        |
> | 20%         | 0.43         | 0.077        |
> | 30%         | 0.45         | 0.083
> | 50%         | 0.47         | 0.095        |
> | 80%         | 0.49         | 0.11         |
>
>
> The results show that prediction errors do affect performance, but the impact is limited. Even at 80% error scale, P99 TTFT remains under 0.5s.

---

> > ### Author Rebuttal · Reviewer_7e2U · 2026-03-31
> >
> > My concerns have been addressed. The authors have added large-scale simulation experiments as well as a sensitivity study on prediction error, which makes the paper more convincing. I will keep my positive score.

---

> > > ### Author Response · Authors · 2026-04-03
> > >
> > > Thank you for carefully reviewing our rebuttal and for confirming that your concerns have been adequately addressed. We sincerely appreciate your constructive and insightful feedback throughout the review process. We will incorporate the simulation results into the revised manuscript.

---

### Decision · Program_Chairs · 2026-04-30

**Decision:**

Accept (regular)

**Comment:**

This paper addresses an important and practical problem in multi-LLM serving and proposes a simple but effective idea: proactively preload weights for multiple models so instances can be brought up quickly when demand spikes. The empirical results are strong, and the reviewers generally agree that the problem is relevant and the system is well motivated.

My main reservation is that the core idea is relatively straightforward, so the novelty comes more from the system design and implementation than from a fundamentally new conceptual insight. The evaluation is also somewhat limited in real-world scale, and some concerns remain around cost analysis and the generality of the simulation results, although the rebuttal improved the case substantially.

A further weakness is reproducibility and practical impact: the submission does not include code release, limiting the framework's immediate usefulness to the community.

Overall, I view this as a borderline but positive submission. The problem matters, the implementation appears substantial, and the latency improvements are compelling.